# Diagnostic accuracy of self-reported food consumption and shaking chills in predicting bacteremia in outpatients: A prospective, multicenter observational study

Takayuki Komatsu[1,2]*, Kenji Inoue[3,4], Yuki Someya[2,5], Hiroshi Hirano[6], Fumihiro Saitoh[7], Makoto Aoki[8], Akihiro Inui[9], Lawrence M. Tierney Jr[10], Kentaro Mishima[1], Keiko Mizuno[1], Hiroki Takami[1], Tomohisa Nomura[1], Manabu Sugita[1]

1 Department of Emergency and Critical Care Medicine, Juntendo University Nerima Hospital, Tokyo, Japan, 2 Department of Sports Medicine, Faculty of Medicine, Juntendo University, Tokyo, Japan, 3 Department of Cardiovascular Biology and Medicine, Juntendo University Nerima Hospital, Tokyo, Japan, 4 Tokyo heart rhythm clinic Shinjuku, Tokyo, Japan, 5 Graduate School of Health and Sports Science, Juntendo University, Chiba, Japan, 6 Department of Surgery, Kouseikai Suzuki Hospital, Tokyo, Japan, 7 Department of General Medicine, Ooizumi Health Cooperative Hospital, Tokyo, Japan, 8 Infectious Disease Consultant, Japan, 9 Department of General Medicine, Faculty of Medicine, Juntendo University, Tokyo, Japan, 10 Department of Internal Medicine, University of California San Francisco, San Francisco, California, United States of America

* tkomatu@juntendo.ac.jp

## Abstract

Bacteremia, a critical condition that can lead to sepsis, is often diagnosed using blood cultures, which may yield false positives, leading to unnecessary treatments. Although clinical indicators, such as shaking chills and food consumption, have been identified as predictors of bacteremia, their diagnostic accuracy in outpatients, particularly when considering the timing of blood collection, remains unclear. This study aimed to assess the diagnostic accuracy of self-reported food consumption and shaking chills in detecting bacteremia, focusing on the time interval between the last meal and blood culture collection. This prospective, multicenter, observational study included outpatients aged > 16 years who could eat orally and underwent blood cultures in the emergency or general medicine department from April 2019 to March 2021. Food consumption before blood culture was self-reported using a medical questionnaire and categorized as "normal" (≥80%) or "poor" (<80%). The presence of chills was also assessed. Among 534 patients (mean age 68.3 ± 21.9 years, 51.3% men), 68 had bacteremia. The absence of poor food consumption (i.e., normal food consumption) had a negative predictive value of 91.2% (95% confidence interval, 88.8–93.6) and a negative likelihood ratio of 0.66 (0.23–1.94). Excluding the blood cultures collected between 10 pm and 8 am, these values increased to 96.2% (94.5–97.8) and 0.32 (0.12–0.89), respectively. Shaking chills had a positive likelihood ratio of 3.74 (2.75–4.73), increasing to 4.21 (3.22–5.19) after the same exclusion. Self-reported shaking chills were good positive predictors of bacteremia in outpatients,

**Data availability statement:** The data that support the findings of this study are not publicly available because of the policy of the Ethics Committee of Juntendo University Nerima Hospital due to the potential concern that participants can be identified based on their individual data. However, the data are available from Masashi Nagao, an independent data manager affiliated with the Clinical Research & Trial Center of Juntendo University (nagao@juntendo.ac.jp), upon reasonable request.

**Funding:** This study was funded by JSPS KAKENHI Grant numbers JP19K16990 and JP23K10725.

**Competing interests:** The authors have declared that no competing interests exist.

whereas self-reported normal food consumption, when accounting for the time between meals, ruled out bacteremia. These findings could help improve the early diagnosis and management of bacteremia, particularly in outpatient settings, and may contribute to the development of self-report tools for clinical decision-making.

## Introduction

Bacteremia, indicated by a true-positive blood culture result reflecting the presence of bacteria in the bloodstream [1], is a major cause of morbidity and mortality [2]. Community-onset bacteremia has an annual incidence of 43–101.2 per 100,000 [3–8]. Although blood cultures (BCs) are commonly used to detect bacteremia, up to half of them may be contaminants, leading to unnecessary treatment [1]. In cases of sepsis or septic shock, early administration of intravenous antibiotics is recommended [9]. Given the potential for bacteremia to rapidly escalate to sepsis, timely and accurate diagnosis is important to minimize patient morbidity and the healthcare burden. However, inappropriate antibiotic use promotes resistance [10,11]. Thus, it is crucial to both rule in and rule out bacteremia using simple clinical data.

In a previous study [12], shaking chills were confirmed as a reliable indicator of bacteremia, with a positive likelihood ratio (+LR) of 4.78, as reviewed by Coburn et al. [1,13,14]. While the absence of systemic inflammatory response syndrome (SIRS) criteria has been considered the best predictor for ruling out bacteremia [1,15], we also found that normal food consumption assessed by nursing staff before BC collection served as a useful negative predictor, with a negative likelihood ratio (–LR) of 0.18 [12] and a negative predictive value (NPV) of 98.3% [16]. However, these findings were not applicable to outpatients. Although significant research has been conducted on hospitalized patients, there remains a gap in outpatient settings that this study aims to address. The maintenance of appetite is commonly considered a good condition, making it a useful clinical indicator in outpatient settings where invasive testing may not be feasible. Additionally, owing to the extended fasting period between dinner and breakfast, appetite may not be accurately reflected based on the timing of BC collection. A prolonged fasting period between dinner and breakfast may distort appetite assessments, leading to inaccurate predictions. While clinical history typically considers timing, no studies have examined bacteremia prediction while accounting for this factor. Thus, this study aimed to evaluate the diagnostic accuracy of self-reported food consumption and shaking chills in predicting bacteremia in outpatients, particularly in relation to the timing of BC collection and the last meal.

## Materials and methods

### Study design and patients

This prospective, multicenter observational study involved three hospitals in Tokyo, Japan: Juntendo University Nerima Hospital (490 beds), which serves both the university and the community as a regional core hospital; and Ooizumi Health

Cooperative Hospital (94 beds) and Kouseikai Suzuki Hospital (99 beds), both of which are community hospitals. Outpatients aged >16 years who underwent BC and completed a medical questionnaire between April 1, 2019, and March 31, 2021, were enrolled. All patients either arrived by ambulance or presented to the emergency, general medicine, or relevant departments.

## Ethics approval

All procedures performed in this study involving human participants were in accordance with the ethical standards of the institutional and/or national research committee, as well as the 1964 Declaration of Helsinki and its later amendments or comparable ethical standards. This study was approved by the Ethics Committee of Juntendo University Nerima Hospital (Approval No. N18-0051) and each participating center. The requirement for written informed consent was waived by the Ethics Committee of Juntendo University Nerima Hospital owing to the use of anonymized data. This study adhered to the Standards for Reporting Diagnostic Accuracy guidelines [17] and was registered in the University Hospital Medical Information Network Clinical Trial Registry (UMIN000036354) (date of registration: 30th/Mar/2019).

## Exclusion criteria

Patients who underwent tube feeding, gastrostomy, or total or peripheral parenteral nutrition were also excluded. Additionally, in a previous study [16], some patients in the bacteremia group maintained normal food consumption regardless of the presence of shaking chills. Therefore, patients or their representatives who provided uncertain information regarding food consumption or chills were excluded to ensure the accuracy of the data. Additionally, patients whose condition had not been checked for > 12 h before BC collection were excluded to ensure an accurate reflection of their status.

## Collecting blood cultures

To avoid interrupting the treatment, BCs were collected at the discretion of the treating physicians based on history taking, physical examination, or any tests consistent with our previous studies [12,16]. After skin decontamination, at least two sets of BCs were collected by the physician, with each set consisting of one aerobic bottle and one anaerobic bottle, with a total blood volume of 8–10 mL per set. The samples were primarily obtained in the supine position from different peripheral sites, including the arteries (e.g., radial or femoral artery) and veins (e.g., cephalic, medial cubital, femoral, internal jugular, and subclavian veins). BD BACTECTM blood culture bottles and BD BACTECTM FX incubation equipment (Becton Dickinson, Sparks, MD, USA) were used at all participating hospitals.

## Definition of true bacteremia and contamination

As in previous studies [12,16,18–20], true bacteremia was defined as an organism isolated from two sets of BCs. Additionally, a single positive BC was considered bacteremia if it matched the clinical presentation. Contaminants were identified through a consensus decision made by the infection control team as part of their daily medical practice in each hospital. This team consisted of at least two investigators, including clinical microbiologists and physicians, as described by Rothe et al. [21] and in previous studies [12,16,22]. Contaminants were defined as organisms common to the skin flora (e.g., *Bacillus* species, coagulase-negative *Staphylococcus*, *Corynebacterium* species, and *Micrococcus* species) that were not isolated from another potentially infected site in a patient with incompatible clinical features and no risk factors for infection with the isolated organism, nor did they show matching antibiotic susceptibilities. Furthermore, when a single positive BC result was detected that did not align with the clinical presentation, it was considered a contaminant. In this study, "bacteremia" refers to true bacteremia, while combined negative BCs and contamination were classified as "non-bacteremia".

### Prediction variables of bacteremia

**Medical questionnaire.** We evaluated food consumption and the degree of chills as described in previous studies [12,16]. As nurses could not assess outpatients, patients or their representatives completed a self-administered medical questionnaire. Upon arrival at the hospital, the respondents selected the appropriate response for each component.

**Food consumption.** Food consumption was categorized based on the meal consumed immediately before (i.e., last meal) BC collection: low (<50%), moderate (≥50% to <80%), and high (≥80%), with an "uncertain" option if the respondent forgot. After collecting the questionnaires, food consumption was grouped as in previous studies [12]: high food consumption, referred to as the "normal food consumption groups," and the integrating low and moderate food consumption, referred to as the "poor food consumption groups."

**Chills.** As in previous studies [12–14], the degree of chills at or immediately before BC collection was categorized into four levels: "no chills" (absence of chills), "mild chills" (feeling cold, needing an outer jacket), "moderate chills" (very cold, needing a thick blanket), and "shaking chills" (extremely cold with generalized shaking, even under a thick blanket). An "uncertain" option was also included if the patient or their caregiver had not remembered or determined the severity of the chills. Following a previous study [12], we categorized patients into "shaking chills groups" and "negative shaking chills groups" (no, mild, or moderate chills).

**Other predictive variables.** Additional predictive variables included age, sex, Glasgow Coma Scale (GCS) score, body temperature (BT), heart rate (HR), systolic blood pressure (SBP), diastolic blood pressure, respiratory rate (RR), white blood cell count (WBC), and C-reactive protein (CRP) level. These were recorded at or immediately before BC collection. The presence of SIRS (≥2 SIRS criteria calculated from BT, HR, RR, and WBC) [23] and quick Sequential Organ Failure Assessment (qSOFA) score, which consists of GCS, SBP, and HR [24], were also assessed. CRP levels were dichotomized as >10.0 or ≤10.0 mg/dL [12,25]. Additionally, we reviewed the method of arrival (ambulance or self-transport), the presence of dementia, prior antibiotic administration, and whether BCs were collected between 10 pm and 8 am due to extended fasting between dinner and breakfast.

### Statistical analysis

Based on our previous study [12], bacteremia rates in the poor and normal food consumption groups were 16.3% and 2.4%, respectively. We calculated a minimum sample size of 252 patients using these figures to achieve 90% power, with an alpha error of 0.05. Continuous variables are presented as mean ± SD or median (IQR), depending on normality (Shapiro–Wilk test). To assess the diagnostic accuracy between the bacteremia group (BG) and non-bacteremia groups (non-BG), we used the unpaired *t*-test or Mann–Whitney *U* test for continuous variables and the chi-square test for categorical variables. Considering the components of the SIRS criteria and qSOFA score, it was challenging to accurately calculate the number of patients with SIRS or qSOFA score ≥2 using the imputed values. Therefore, after excluding cases with missing data, the association between any variables and bacteremia was assessed using multiple logistic regression analysis with a level of statistical significance ($P < 0.05$) for the covariates of sex, CRP, shaking chills, and SIRS, which were identified in previous studies [1,12]. In addition, age, food consumption, qSOFA score, method of arrival, and time interval of collecting BCs were also analyzed as covariates in this study. We calculated the sensitivity, specificity, positive predictive value (PPV), NPV, and likelihood ratios (+LR and –LR) for bacteremia prediction. A subgroup analysis was performed, excluding patients whose BC samples were collected between 10 pm and 8 am. All statistical tests were two-tailed ($P < 0.05$) and performed using SPSS for Windows (version 28.0; IBM Corp., Armonk, NY, USA) by a physician (KT) and an independent statistician (SY).

## Results

### Patient characteristics

After excluding 181 patients, 534 patients (mean age 68.3 ± 21.9 years, 51.3% men) were included, with 68 (12.7%) in the BG and 466 (including 21 [3.9%] contamination cases) in the non-BG (Fig 1). Baseline patient characteristics are

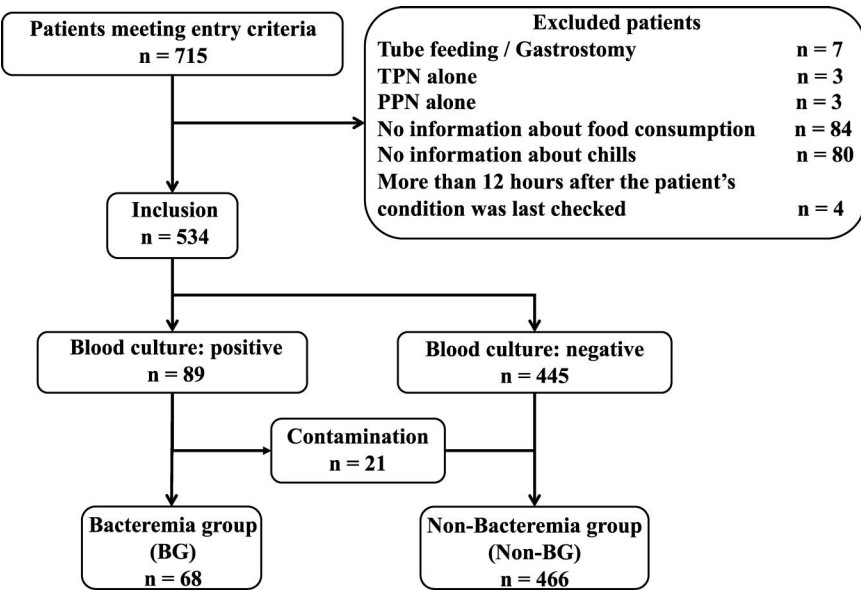

**Fig 1. Study population.** TPN, total parenteral nutrition; PPN, peripheral parenteral nutritional; BG, bacteremia group.

presented in Table 1. Patients in the BG were significantly older (median [IQR]: 83.0 [77.0–87.0] *vs.* 74.0 [49.0–84.0]; *P*<0.001), although the incidence of dementia did not differ. More patients in the BG arrived by ambulance (53 [77.9%] *vs.* 214 [45.9%]; *P*<0.001) and had BCs collected between 10 pm and 8 am (19 [27.9%] *vs.* 71 [15.2%]; *P*<0.001). BT, HR, and RR were significantly higher in the BG (*P*<0.001). The number of patients with GCS ≥ 14 was lower in the BG (82.4% *vs.* 90.5%; *P*=0.001), despite having the same median GCS and IQR. Although SBP was not statistically different, more patients in the BG had an SBP ≤ 100 mmHg (*P*=0.019). Patients with a qSOFA score ≥2 were more frequent in the BG (*P*=0.001). CRP levels were significantly higher in the BG (11.0 [2.7–17.8] *vs.* 5.3 [1.1–11.9]; *P*=0.001), and although there was no statistical difference in WBC count, more patients in the BG met the SIRS criteria (*P*<0.001). Furthermore, a smaller number of patients in the BG were administered antibiotics before their arrival (4 [5.9%] *vs.* 72 [15.5%]; *P*<0.035).

## Pathogen and diagnosis

The main pathogens causing bacteremia were *Escherichia coli* (*n*=21, 30.1%), *Klebsiella pneumoniae* (*n*=12, 17.6%), and *Streptococcus pneumoniae* (*n*=5, 7.4%) (Table 2). Other pathogens include *Streptococcus* sp., Enterobacterales, Anaerobes sp., and *Staphylococcus* sp. Primary clinical diagnoses of BG were pyelonephritis, cholangitis, abscess, pneumonia, cellulitis, bacterial meningitis, osteomyelitis, infective endocarditis, cholecystitis, and intrauterine infection (Table 3).

## Diagnostic accuracy

**Food consumption.** The poor and normal food consumption groups comprised 432 and 102 patients, respectively (Table 1). Among them, 59 (13.7%) and nine (8.8%) patients had bacteremia, respectively. Poor food consumption showed a sensitivity of 86.8% (95% confidence interval [CI], 83.9–89.6) for predicting bacteremia. Conversely, the absence of poor food consumption (i.e., normal food consumption) had an NPV of 91.2% (95% CI, 88.8–93.6) and –LR of 0.66 (95% CI, 0.23–1.94) (Table 4). Of the nine patients with normal food consumption in the BG, six had BCs collected between 10 pm and 8 am. One patient had an abscess, another had an indwelling bladder catheter with pyelonephritis and hematuria, and one was diagnosed with cellulitis but was readmitted >24 h after discharge due to fever without BC

**Table 1. Patient characteristics.**

| | Bacteremia | Non-bacteremia | P-value |
|---|---|---|---|
| | (*n* = 68) | (*n* = 466) | |
| Age (years) | 83.0 (77.0–87.0) | 74.0 (49.0–84.0) | <0.001 |
| <65 | 7 (10.3%) | 170 (36.5%) | <0.001 |
| 65–74 | 9 (13.2%) | 65 (13.9%) | 0.874 |
| 75–84 | 24 (35.3%) | 116 (24.9%) | 0.068 |
| ≥85 | 28 (41.2%) | 115 (24.7%) | 0.004 |
| No. (%) of women | 34 (50.0%) | 226 (48.5%) | 0.817 |
| No. (%) of patients who visited the hospital by | | | |
| Ambulance | 53 (77.9%) | 214 (45.9%) | <0.001 |
| Themselves | 15 (22.1%) | 252 (54.1%) | <0.001 |
| No. (%) of patients collected BCs from 10 pm to 8 am | 19 (27.9%) | 71 (15.2%) | 0.009 |
| No. (%) of patients administered antibiotics before arrival | 4 (5.9%) | 72 (15.5%) | 0.035 |
| No. (%) of patients with dementia | 10 (20.6%) | 60 (12.9%) | 0.086 |
| Vital signs | | | |
| Axial body temperature; BT (ºC) | 38.4 ± 1.3 | 37.5 ± 1.0 | <0.001 |
| Glasgow Coma Scale; GCS | 15.0 (14.0–15.0) | 15.0 (14.0–15.0) | 0.001 |
| GCS ≥ 14 | 56 (82.4%) | 420 (90.5%) | 0.040 |
| Heart rate; HR (bpm) | 107.9 ± 20.4 | 95.4 ± 19.1 | <0.001 |
| Systolic blood pressure; SBP (mmHg) | 131.9 ± 28.2 | 131.3 ± 26.4 | 0.916 |
| Diastolic blood pressure; DBP (mmHg) | 70.5 ± 15.3 | 73.5 ± 15.7 | 0.327 |
| Respiratory rate; RR (/min) | 20.0 (18.0–22.5) | 18.0 (16.0–22.0) | 0.003 |
| Laboratory data | | | |
| White blood cell counts; WBC (×10³ WBC/μL) | 10.5 (7.1–14.9) | 9.2 (6.6–13.2) | 0.259 |
| C-reactive protein; CRP (mg/dL) | 11.0 (2.7–17.8) | 5.3 (1.1–11.9) | 0.001 |
| CRP > 10 mg/dL, No. (%) | 37 (54.4%) | 146 (31.5%) | <0.001 |
| No. (%) with a qSOFA score ≥2 | 22 (33.3%) | 68 (16.2%) | 0.001 |
| GCS < 15, No. (%) | 30 (44.1%) | 118 (25.4%) | 0.001 |
| RR ≥ 22/min, No. (%) | 26 (40.0%) | 115 (29.8%) | 0.101 |
| SBP ≤ 100 mmHg, No. (%) | 16 (23.9%) | 57 (13.1%) | 0.019 |
| No. (%) with SIRS (≥2 components of SIRS criteria) | 56 (84.8%) | 233 (55.7%) | <0.001 |
| HR > 90 bpm, No. (%) | 56 (83.6%) | 273 (62.5%) | 0.001 |
| RR > 20/min, No. (%) | 40 (61.5%) | 147 (38.1%) | <0.001 |
| BT < 36 ºC or >38 ºC, No. (%) | 43 (63.2%) | 156 (34.1%) | <0.001 |
| WBC < 4,000 × 10³ WBC/μL or >12,000 × 10³ WBC/μL, No. (%) | 32 (47.1%) | 159 (34.2%) | 0.039 |
| Degree of chills, No. (%) | | | |
| Shaking chills | 18 (26.5%) | 33 (7.1%) | <0.001 |
| Negative shaking chills | 50 (73.5%) | 433 (92.9%) | <0.001 |
| Moderate chills | 11 (16.2%) | 40 (8.6%) | 0.047 |
| Mild chills | 12 (17.6%) | 73 (15.7%) | 0.676 |
| No chills | 27 (39.7%) | 320 (68.7%) | <0.001 |
| Amount of food consumption, No. (%) | | | |
| High (normal food consumption) | 9 (13.2%) | 93 (20.0%) | 0.188 |
| Not high (poor food consumption) | 59 (86.8%) | 373 (80.0%) | 0.188 |
| Moderate | 12 (17.6%) | 61 (13.1%) | 0.307 |
| Low | 47 (69.1%) | 312 (67.0%) | 0.722 |

Data are expressed as the mean ± SD, median (IQR), or n (%). *P*-value: unpaired *t*-test, Mann–Whitney *U* test, or chi-square test, residual analysis. BC, blood culture; qSOFA, quick Sequential Organ Failure Assessment; SIRS, systemic inflammatory response syndrome.

**Table 2. Pathogens identified in patients with true bacteremia.**

| Organisms | No. | Organisms | No. |
|---|---|---|---|
| **Single pathogen identified** | | | |
| *Escherichia coli* | 17 | *Bacteroides fragilis* | 1 |
| *Klebsiella pneumoniae* | 9 | *Peptostreptococcus sp* | 1 |
| *Streptococcus pneumoniae* | 5 | *Streptococcus intermedius* | 1 |
| ESBL-producing *Escherichia coli* | 4 | *Streptococcus oralis* | 1 |
| *Group G streptococcus* | 4 | *Group A streptococcus* | 1 |
| *Staphylococcus epidermidis* | 3 | *Streptococcus gallolyticus* | 1 |
| *Group B streptococcus* | 3 | *Staphylococcus capitis subsp. capitis* | 1 |
| *Staphylococcus aureus* | 2 | ESBL-producing *Klebsiella pneumoniae* | 1 |
| *Proteus mirabilis* | 2 | | |
| **Two pathogens identified from one blood culture** | | | |
| *Staphylococcus aureus* Coagulase Negative *Staphylococci* | 1 | *Prevotella/Porphyromonas Propionibacterium sp* | 1 |
| *Microaetophilic streptococcus Staphylococcus capitis subsp ureolyticus* | 1 | *Klebsiella pneumoniae Citrobacter freundii* | 1 |
| *Proteus mirabillis Group B streptococcus* | 1 | *Proteus mirabilis Escherichia coli* | 1 |
| *Morganella morganii Enterococcus faecalis* | 1 | *Eubacterium sp Prevotella/Porphyromonas* | 1 |
| *Escherichia coli Staphylococcus lugdunensis* | 1 | *Escherichia coli Streptococcus constellatus* | 1 |
| **Three pathogens identified from one blood culture** | | | |
| *Pseudomonas aeruginosa, Enterococcus avium, Bacteroides fragilis group* | | | 1 |

ESBL: Extended-spectrum beta-lactamase.

collection (Table 5). After excluding patients with BCs collected between 10 pm and 8 am, sensitivity, NPV, and –LR improved to 93.8% (95% CI, 91.8–95.9), 96.2% (95% CI, 94.5–97.8), and 0.32 (95% CI, 0.12–0.89), respectively (Table 4).

**Chills.** The negative shaking-chills and shaking-chills groups included 483 and 51 patients, respectively (Table 1). Among them, 50 (10.4%) and 18 (35.3%) patients had bacteremia, respectively. Shaking chills had a specificity of 92.9% (95% CI, 90.7–95.1), NPV of 89.6% (95% CI, 87.1–92.2), and +LR of 3.74 (95% CI, 2.75–4.73) for predicting bacteremia (Table 2). After excluding certain patients, the specificity, NPV, and +LR improved to 94.2% (95% CI, 92.2–96.2), 91.0% (95% CI, 88.5–93.4), and 4.21 (95% CI, 3.22–5.19), respectively (Table 4).

**Other predictive variables.** Similarly, the presence of qSOFA score ≥2, SIRS, and CRP > 10 mg/dL had a +LR of 2.06 (95% CI, 1.07–3.06), –LR of 0.34 (95% CI, 0.12–0.94), and NPV of 91.1% (95% CI, 88.7–93.5) for predicting bacteremia. However, the diagnostic accuracy of these variables did not improve after excluding the same patients (Table 4).

## Ruling in bacteremia

Multiple logistic regression analysis showed that the significant predictors of bacteremia were shaking chills (odds ratio [OR] 3.20; 95% CI, 1.53–6.67; *P* = 0.002), age ≥ 75 years (OR 2.89; 95% CI, 1.41–5.90; *P* = 0.004), CRP > 10.0 mg/dL (OR 2.74; 95% CI, 1.50–4.98; *P* = 0.001), SIRS (OR, 2.75; 95% CI, 1.28–5.54; *P* = 0.010), and transportation by ambulance (OR 2.46; 95% CI, 1.17–5.18; *P* = 0.017) (Table 6). After excluding patients with BCs collected between 10 pm and 8 am, the OR for shaking chills and age ≥ 75 years increased to 4.34 (95% CI, 1.79–10.50; *P* = 0.001) and 3.83 (95% CI, 1.74–8.44; *P* = 0.001), respectively.

**Table 3. Definitive diagnosis of study patients.**

|  | Bacteremia (n = 68) | Non-Bacteremia (n = 466) |
|---|---|---|
| **Infectious disease** | | |
| Pyelonephritis | 22 | 50 |
| Cholangitis | 8 | 6 |
| Abscess | 6 | 16 |
| Pneumonia | 4 | 92 |
| Cellulitis | 3 | 17 |
| Catheter-related bloodstream infection | 2 | |
| Febrile neutropenia | 2 | 7 |
| Bacterial meningitis | 2 | |
| Osteomyelitis | 1 | 3 |
| Infective endocarditis | 1 | |
| Infective aortic abdominal aneurysm | 1 | |
| Infective portal vein embolism | 1 | |
| Cholecystitis | 1 | 7 |
| Diabetic gangrene | 1 | |
| Gastroenteritis | 1 | 9 |
| Intrauterine infection | 1 | |
| Purulent arthritis | 1 | |
| Source of unknown origin | 10 | |
| Upper respiratory tract infection | | 26 |
| Appendicitis | | 9 |
| Peritonitis | | 8 |
| Infectious mononucleosis | | 6 |
| Bacterial tonsillitis | | 5 |
| Viral meningitis | | 3 |
| Prostatitis | | 3 |
| Surgical site infection | | 2 |
| Necrotizing fasciitis | | 1 |
| Ludwig angina | | 1 |
| Pelvic inflammatory disease | | 1 |
| Other infection | | 9 |
| **Non-infectious disease** | | |
| Connective tissue disease | | 24 |
| Gastrointestinal disease | | 24 |
| Malignant-associated fever | | 12 |
| Cardiovascular disease | | 11 |
| Metabolic and endocrine disease | | 10 |
| Pulmonary disease | | 8 |
| Periodic febrile illness | | 4 |
| Heat stroke | | 4 |
| Dermatological disease | | 3 |
| Cerebrovascular disease | | 2 |
| Orthopedic disease | | 2 |
| Ureteral stone attack | | 2 |

*(Continued)*

| | Bacteremia (n = 68) | Non-Bacteremia (n = 466) |
|---|---|---|
| Drug adverse effects | | 2 |
| Rhabdomyolysis | | 1 |
| Focus unknown | | 81 |

The sites of abscess in the BG were as follows: two subcutaneous abscesses at the buttock and one each in the liver, prostate, iliopsoas, and left thigh (intramuscular). In the non-BG, the sites included four subcutaneous abscesses, three intraperitoneal, two in the liver, and one each in the kidney, iliopsoas, intrauterine, intragingival, peritonsillar, perianal, and urachal remnants.

## Discussion

This study found that poor self-reported food consumption had a sensitivity of 86.8% for predicting bacteremia; conversely, the absence of poor food consumption (i.e., self-reported normal food consumption) had an NPV of 91.2% and −LR of 0.66. After excluding patients with BCs collected between 10 pm and 8 am, the sensitivity, NPV, and −LR improved to 93.9%, 96.2%, and 0.32, respectively. Similarly, shaking chills had a specificity of 92.9% and a +LR of 3.74 for predicting bacteremia. Excluding these patients increased the specificity and +LR to 94.2% and 4.21, respectively.

Although Takada et al. [26] reported the ineffectiveness of self-reported food consumption in outpatients, with a sensitivity of 84.4%, NPV of 86.8%, and −LR of 0.80, for predicting bacteremia, their study's definition of food consumption ("during the past 24 hours") differed significantly from ours ("immediately prior to BC") [12,16,26], which may explain the discrepancies in the results.

Conversely, the specificity and +LR of shaking chills for predicting bacteremia were slightly lower than those in our previous study [12] (95.1% and 4.78, respectively) or the study by Takada et al. [26] (94.7% and 4.3, respectively). While some studies [13,14,26] used chills within 24 h before presentation as the definition, we defined chills as those occurring immediately before BC collection to align with the timing of food consumption [12]. Additionally, considering a meta-analysis [27] with a specificity of 0.87, and a prospective multicenter cohort study [28] reporting an OR of 5.9 for bacteremia prediction, the presence of shaking chills remains a strong positive predictor, even in outpatients, despite not collecting BCs at the onset of chills, as in hospitalized settings.

This study also confirmed that a qSOFA score ≥2 did not predict bacteremia, as reported in recent studies [29,30]. SIRS had an NPV of 94.9%, which, although not as strong as that reported by Jones and Lowes [15], was better than that reported in other studies [31]. SIRS was also identified as a strong predictor of bacteremia, with an OR of 2.66 in this study. Notably, 65 patients were excluded from the multiple logistic regression analysis because of missing vital signs, including the SIRS and qSOFA score. However, except for one patient, these individuals were not transported by ambulance and were likely not critically ill, which suggests that the additional contribution of SIRS and qSOFA score in predicting bacteremia was minimal.

The strength of this study lies in the use of direct patient information as a reliable tool for predicting bacteremia in outpatients, particularly considering the interval between the last meal and arrival at the hospital. Recent models, such as the modified Shapiro rule with an NPV of 96.8% [31] or new prediction models with a sensitivity of 0.93 [32], are retrospective and difficult to apply to outpatients because of their reliance on blood examinations. To the best of our knowledge, no previous prospective studies have evaluated self-reported food consumption and shaking chills while considering the time course, as physicians do during history-taking. Preventing delays from first pre-hospital contact to hospital admission is crucial, as this is a significant risk factor for 30-day all-cause mortality associated with community-onset bacteremia [33]. The results of this study may contribute to developing a self-screening tool to help determine the necessity of hospital visits, as well as to enhance outpatient medical questionnaires in hospitals.

**Table 4. Diagnostic accuracy for predicting bacteremia.**

| | ALL | | | | | | | Exclusion from 10 pm to 8 am the next day | | | | | |
| | Sensitivity | Specificity | PPV | NPV | +LR | –LR | | Sensitivity | Specificity | PPV | NPV | +LR | –LR |
| --- | --- | --- | --- | --- | --- | --- | --- | --- | --- | --- | --- | --- | --- |
| Poor food consumption | 86.8 (83.9–89.6) | 20.0 (16.6–23.3) | 13.7 (10.7–16.6) | 91.2 (88.8–93.6) | 1.08 (0.09–2.08) | 0.66 (0.23–1.94) | | 93.9 (91.8–95.9) | 19.0 (15.7–22.3) | 12.6 (9.8–15.4) | 96.2 (94.5–97.8) | 1.16 (0.16–2.16) | 0.32 (0.12–0.89) |
| qSOFA score ≥2 | 33.3 (29.1–37.5) | 83.8 (80.6–87.1) | 24.4 (20.6–28.3) | 88.9 (86.1–91.7) | 2.06 (1.07–3.06) | 0.80 (0.29–2.19) | | 31.9 (27.8–36.1) | 84.6 (81.4–87.8) | 21.7 (18.1–25.4) | 90.2 (87.6–92.9) | 2.07 (1.08–3.06) | 0.81 (0.29–2.20) |
| SIRS | 84.8 (81.7–88.0) | 44.3 (39.8–48.7) | 19.4 (15.9–22.9) | 94.9 (92.9–96.8) | 1.52 (0.53–2.51) | 0.34 (0.12–0.94) | | 80.9 (77.3–84.4) | 47.3 (42.8–51.7) | 17.2 (13.8–20.6) | 94.8 (92.8–96.8) | 1.53 (0.55–2.52) | 0.41 (0.14–1.17) |
| Shaking chills | 26.5 (22.7–30.2) | 92.9 (90.7–95.1) | 35.3 (31.2–39.3) | 89.6 (87.1–92.2) | 3.74 (2.75–4.73) | 0.79 (0.29–2.18) | | 24.5 (20.8–28.1) | 94.2 (92.2–96.2) | 34.3 (30.3–38.3) | 91.0 (88.5–93.4) | 4.21 (3.22–5.19) | 0.80 (0.29–2.19) |
| CRP >10mg/dL | 54.4 (50.2–58.6) | 68.5 (64.6–72.5) | 20.2 (16.8–23.6) | 91.1 (88.7–93.5) | 1.73 (0.72–2.74) | 0.67 (0.24–1.84) | | 59.2 (55.0–63.4) | 65.9 (61.9–69.9) | 17.8 (14.5–21.0) | 92.8 (90.6–95.0) | 1.74 (0.72–2.75) | 0.62 (0.23–1.70) |

+LR, positive likelihood ratio; –LR, negative likelihood ratio; NPV, negative predictive value; PPV, positive predictive value; qSOFA, quick Sequential Organ Failure Assessment; SIRS, systemic inflammatory response syndrome.

**Table 5. Further information about patients in the bacteremia group with normal food consumption.**

| | Sex | Age (years) | BCs collected from 10 pm to 8 am | Identified pathogen | Definitive diagnosis | Background |
|---|---|---|---|---|---|---|
| 1 | M | 49 | No | *Staphylococcus aureus* | Intramuscular abscess in the left thigh | Regular hemodialysis. |
| 2 | M | 82 | Yes | *Escherichia coli* | Cholangitis | Sudden onset with shivering 4 hours after the last meal due to obstruction of the bile duct stent for cholangiocarcinoma. |
| 3 | M | 85 | No | *Morganella morganii, Enterococcus feacalis* | Pyelonephritis | Sudden onset when a bladder catheter is placed for BPH and NB. |
| 4 | F | 80 | Yes | *Escherichia coli* | Cholangitis | Nursing home staff found the patient with dyspnea at 4 am after being stable; it took over 3 hours to transport her to the hospital. |
| 5 | F | 83 | Yes | *Group B streptococcus* | Focus unknown | Sudden onset with shivering 2 hours after last meal (10 pm), undergoing chemotherapy for lung carcinoma. |
| 6 | F | 84 | Yes | ESBL-producing *Escherichia coli* | Pyelonephritis | Sudden onset with shivering 8 hours after going to bed at 8 pm. |
| 7 | F | 90 | Yes | *Escherichia coli* | Pyelonephritis | Sudden onset with shivering at 4 am after a normal dinner. |
| 8 | F | 90 | No | *Group G streptococcus* | Cellulitis | Returned to hospital after over 24 hours, initially presented with fever; however, no BCs collected |
| 9 | F | 92 | Yes | *Escherichia coli* | Pyelonephritis | BCs were collected about 14 h after the last dinner |

BC: Blood culture, M: Male, F: Female, ESBL: Extended-spectrum beta-lactamase, BPH: Benign prostate hypertrophy, NB: Neurogenic bladder dysfunction.

**Table 6. Components of predicting bacteremia identified by multiple logistic regression analysis.**

| | ALL (*n*=469) | | Exclusion from 10 pm to 8 am the next day (*n*=379) | |
|---|---|---|---|---|
| Variable | OR (95% CI) | *P*-value | OR (95% CI) | *P*-value |
| Age ≥ 75 years | 2.89 (1.41–5.90) | 0.004 | 3.83 (1.74–8.44) | 0.001 |
| Transportation by ambulance | 2.46 (1.17–5.18) | 0.017 | – | – |
| SIRS | 2.75 (1.28–5.54) | 0.010 | 2.49 (1.08–5.72) | 0.032 |
| CRP > 10 mg/dL | 2.74 (1.50–4.98) | 0.001 | 2.04 (1.03–4.02) | 0.040 |
| Shaking chills | 3.20 (1.53–6.67) | 0.002 | 4.34 (1.79–10.50) | 0.001 |

Sixty-five patients were excluded from the multiple logistic regression analysis due to missing data. Other covariates included sex, blood cultures collected between 10 pm and 8 am, poor food consumption, and a qSOFA score ≥2. CI, confidence interval; CRP, C-reactive protein; OR, odds ratio; qSOFA, quick Sequential Organ Failure Assessment; SIRS, systemic inflammatory response syndrome.

This study had some limitations. First, 70% of the patients in the BG were transported by ambulance. A subgroup analysis by transportation route is warranted because these patients often present with poor conditions. However, the number of samples in each subgroup was insufficient for analysis. Second, physicians' awareness of the predictors may have influenced their decision to collect BC. However, the prevalence of bacteremia in this study was 12.7%, which is nearly the same as the 12.0% reported in our previous study using the same criteria for collecting BC [12]. Additionally, another study conducted in the emergency department reported a 14.3% prevalence of bacteremia [34], suggesting that the decision to collect BC in this study was appropriate. Third, the applicability of these results to other countries may be limited because of the differences in cultural habits and medical systems, particularly regarding ambulance use. Although this study reflects the general situation of Japanese outpatients, as evidenced by the identified pathogens, diagnoses, and typical patterns of hospital attendance, further investigation is required to verify these findings. Fourth, a total of 164 patients were excluded due to incomplete questionnaire responses, despite the sufficient sample size. Additionally, the validity of our findings in patients with dementia remains

unclear due to the small number of cases. However, in clinical settings, family members or caregivers typically monitor a patient's condition, as in this study, making separate evaluations unnecessary, as in the present study. Until further studies are conducted, it is believed that this approach should be discouraged for patients, particularly those with dementia, disturbances in consciousness, or who live without a representative, due to the difficulty in responding to medical questionnaires. Moreover, to increase its utility in clinical settings, proper education of patients and their representatives regarding the recognition of food consumption and shaking chills is necessary. Finally, it was difficult to adapt self-reported food consumption to patients whose BCs were collected between 10 pm and 8 am. Among these 90 patients, 19 of the 65 transported by ambulance had bacteremia, compared to only one of the 25 who self-presented. This patient with cholangitis due to bile duct stent obstruction experienced sudden shivering 4 h after the last meal (Table 5). Additionally, 8.8% (9/102) of patients with normal food consumption had bacteremia. However, as shown in Table 5 and in previous studies [12], cases involving abscesses (including infectious endocarditis), sudden onset, or anatomical and structural risks that predispose individuals to bacteremia should be carefully evaluated. Given the high pre-test probability of bacteremia based on additional information, physicians should always suspect bacteremia in similar scenarios.

## Conclusions

In conclusion, this study of outpatients demonstrated that the presence of shaking chills indicated bacteremia, whereas self-reported normal food consumption effectively ruled out bacteremia when considering the time interval between meals. These findings can aid physicians in clinical decision-making and serve as potential self-reporting tools for patients. Based on our findings, further research is needed to explore the use of this self-report tool in pre-hospital settings.

## Acknowledgments

We thank Editage (www.editage.com) for their assistance with editing and proofreading the manuscript.

## Author contributions

**Conceptualization:** Takayuki Komatsu, Kenji Inoue, Manabu Sugita.

**Data curation:** Takayuki Komatsu.

**Formal analysis:** Takayuki Komatsu.

**Funding acquisition:** Takayuki Komatsu.

**Investigation:** Takayuki Komatsu, Yuki Someya, Hiroshi Hirano, Fumihiro Saitoh, Akihiro Inui, Kentaro Mishima, Keiko Mizuno, Hiroki Takami, Tomohisa Nomura, Manabu Sugita.

**Methodology:** Takayuki Komatsu, Kenji Inoue.

**Project administration:** Takayuki Komatsu.

**Supervision:** Kenji Inoue, Manabu Sugita.

**Visualization:** Takayuki Komatsu.

**Writing – original draft:** Takayuki Komatsu.

**Writing – review & editing:** Kenji Inoue, Yuki Someya, Hiroshi Hirano, Fumihiro Saitoh, Makoto Aoki, Akihiro Inui, Lawrence M. Tierney Jr, Kentaro Mishima, Keiko Mizuno, Hiroki Takami, Tomohisa Nomura, Manabu Sugita.

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
