## [Decision Letter · Decision Letter 0]

Dear Dr. Komatsu,

Thank you for submitting your manuscript to PLOS ONE. After careful consideration, we feel that it has merit but does not fully meet PLOS ONE’s publication criteria as it currently stands. Therefore, we invite you to submit a revised version of the manuscript that addresses the points raised during the review process.

We look forward to receiving your revised manuscript.

Kind regards,

Adriana Calderaro

Academic Editor

PLOS ONE

Journal Requirements:

“This study was funded by JSPS KAKENHI Grant-in-Aid for Early-Career Scientists (Grant number JP19K16990)”

3. In this instance it seems there may be acceptable restrictions in place that prevent the public sharing of your minimal data. However, in line with our goal of ensuring long-term data availability to all interested researchers, PLOS’ Data Policy states that authors cannot be the sole named individuals responsible for ensuring data access (http://journals.plos.org/plosone/s/data-availability#loc-acceptable-data-sharing-methods ).

Reviewers' comments:

Reviewer's Responses to Questions

**Comments to the Author**

1. Is the manuscript technically sound, and do the data support the conclusions?

Reviewer #1: Partly

Reviewer #2: Yes

2. Has the statistical analysis been performed appropriately and rigorously?

Reviewer #1: No

Reviewer #2: Yes

3. Have the authors made all data underlying the findings in their manuscript fully available?

Reviewer #1: Yes

Reviewer #2: Yes

4. Is the manuscript presented in an intelligible fashion and written in standard English?

Reviewer #1: Yes

Reviewer #2: Yes

Reviewer #1: I would like to commend the authors for conducting this valuable study and for the opportunity to review it. Developing simple and practical algorithms to predict bacteremia, especially patient-friendly ones, is crucial for reducing sepsis-related mortality and morbidity. This article evaluates the utility of “self-reported” food consumption and shaking chills immediately before taking blood cultures (BC) in predicting the presence or absence of bacteremia in outpatients, primarily those arriving by ambulance. The findings suggest that these self-reported factors are useful predictors of bacteremia in this population. The study’s objectives and clinical implications are clear, and its results hold potential relevance for both patients and practitioners. However, revisions are necessary to enhance the scientific rigor and ensure readers can interpret the findings appropriately.

Major comments

1. Inclusion criteria

The authors assessed the diagnostic accuracy of food consumption and shaking chills separately. It would be helpful to clarify why patients who reported only food consumption or shaking chills were excluded from the analysis. Please provide the rationale for this decision and discuss how it might impact the generalizability of the findings.

2. Potential confounding

If available, please provide data on the time elapsed between symptom onset and BC collection. Since “food consumption immediately before BC collection” could be influenced by this time interval, understanding its impact on the study results is critical.

3. Impact of predictors on BC collection decisions

Physicians’ awareness of the predictors (e.g., food consumption and shaking chills) could have influenced their decision to collect BC. If this information was accessible to physicians before BC collection, it may introduce bias. To address this, the authors might consider discussing whether the prevalence of bacteremia observed in this study (12.7%) aligns with rates reported in similar settings. This comparison could help assess the potential bias in the study results.

4. Multiple logistic regression models

To assess the diagnostic performance of food consumption and shaking chills, it would be appropriate to use multiple logistic regression models that adjust for potential confounding variables. Instead of employing forward or backward stepwise selection, variable inclusion should be guided by a conceptual model outlining plausible confounders. Additionally, given that over 10% of patients were excluded from the analysis due to missing data, the authors should address the potential impact of these exclusions. Performing sensitivity analyses using imputation methods to handle missing data could strengthen the robustness of the findings.

Minor comments

5. Guidelines

Since the article focuses on diagnostic performance, adherence to the STARD guidelines would be more appropriate than STROBE. Please consider revising the reporting accordingly.

6. Interrater reliability for the assessment of contamination

The assessment of blood culture contamination is an important aspect of the study. Please provide data on the interrater reliability of this assessment to ensure the consistency and validity of these findings.

Reviewer #2: Thank you for the opportunity to review this study. The authors have investigated the relationship between dietary intake and the positivity rate of blood cultures. I would like to point out several concerns to improve the quality of this study.

Major comments

#1 Abstract: Even if a blood culture is a false positive, it is not difficult to determine contamination, and false negatives have a greater impact. Therefore, efforts should be made to reduce false negatives. Poor food consumption had a negative likelihood ratio of 0.66 (95% CI: 0.23-1.94). As the authors wrote it in the limitation, the food consumption was only useful for blood cultures collected between 10 pm and 8 am. A negative predictive value between 8 am and 10 pm was 0.32 (0.12-0.89). This figure was not so low, and it is insufficient to exclude bacteremia.

#2 Study design: Please list the names of the three hospitals that participated in the study. Are they all university hospitals? The fact that only 715 blood cultures were ordered in two years is significantly lower compared to community hospitals. Are there few patients who visit without an appointment due to fever?

#3 Collecting blood cultures: How many sets of blood cultures were collected and from which sites? Is there only one case of contamination with S. capitis? If so, the contamination rate seems very low.

#4 Table 6: Why were prior antibiotic use, viral infections, and non-infectious diseases not included as explanatory variables in the multivariate analysis? These clearly affect the positivity rate of blood cultures and should be adjusted for as confounding factors.

#5 Discussion: If dietary intake is low, the negative predictive value of blood cultures is high at 91%, but if dietary intake is not low, blood cultures are less likely to be positive. To demonstrate this, it would be easier to understand if the authors show that the positive predictive value of blood cultures is low when dietary intake is adequate.

#6 Discussion: The importance of chills is the same for both outpatient and inpatient settings. In a university hospital patient group, there are fewer elderly patients and dementia patients, making it difficult to ascertain dietary intake and the presence of chills. Therefore, these research findings may not be applicable to community hospitals. It would be better to add this as a limitation.

#7 Discussion: Even if dietary intake is adequate, nearly 10% (9/102) still had positive blood cultures. Therefore, even with adequate dietary intake, blood cultures should not be omitted. I am concerned that if you decide not to collect blood cultures from patients who can eat, it may delay the diagnosis of infections that can only be diagnosed through blood cultures, such as infective endocarditis, thereby increasing the risk of poor outcomes for such patients.

Minor comments:

#1 Chills: How was "uncertain" classified for chills?

#2 Pathogen and diagnosis: Enterobacteriaceae is now referred to as Enterobacterales.

#3 Pathogen and diagnosis: Which site does the abscess refer to?

#4 Table 2: Only a part of it is labeled as beta-strept. The "beta" is unnecessary.

**Do you want your identity to be public for this peer review?** For information about this choice, including consent withdrawal, please see our Privacy Policy

Reviewer #1: No

Reviewer #2: **Yes: ** Tomohiro Taniguchi

---

## [Author Response · Author response to Decision Letter 1]

28 Feb 2025

February 28, 2025

Professor Adriana Calderaro

Academic Editor

PLOS ONE

Dear Professor Calderaro:

Thank you very much for your review of our manuscript entitled “Diagnostic accuracy of self-reported food consumption and shaking chills in predicting bacteremia in outpatients: a prospective, multicenter observational study”. First, we apologize for the insufficient points in the journal requirements. We edited our manuscript, including those points. Moreover, we appreciate the comments provided and, in response to these comments, have revised the manuscript PONE-D-24-58497 as much as we could.

The following is a point-by-point reply to the Journal Requirements and the reviewer’s comments as follows:

We hope that we have adequately addressed the concerns of the editor and reviewers. We believe that the manuscript is significantly improved and hope that you share our enthusiasm. We look forward to a publication of our manuscript in PLOS ONE.

Sincerely yours,

Takayuki Komatsu

Department of Emergency and Critical Care Medicine,

Juntendo University Nerima Hospital, Tokyo, Japan,

Postal address: 2-1-1 Hongo, Bunkyo-ku,

Tokyo 113-8421, Japan

Tel.: +81-3-5802-1937;

Fax: +81-3-5802-1937;

Email: tkomatu@juntendo.ac.jp;

ORCID: 0000-0002-8730-2081

Journal Requirements:

Comment 1) Please ensure that your manuscript meets PLOS ONE’s style requirements, including those for file naming.

Reply 1: Thank you very much for your comment. We have rechecked the PLOS ONE style templates and revised the manuscript accordingly.

Comment 2: Please state what role the funders took in the study. If the funders had no role, please state: “The funders had no role in study design, data collection and analysis, decision to publish, or preparation of the manuscript.” If this statement is not correct you must amend it as needed. Please include this amended Role of Funder statement in your cover letter; we will change the online submission form on your behalf.

Reply 2: Thank you very much for your suggestion. We have added the following sentences at the end of the revised cover letter: “This study was funded by JSPS KAKENHI Grant-in-Aid for Early-Career Scientists (Grant number JP19K16990). The funders had no role in study design, data collection and analysis, decision to publish, or preparation of the manuscript.”

Comment 3) In this instance it seems there may be acceptable restrictions in place that prevent the public sharing of your minimal data. However, in line with our goal of ensuring long-term data availability to all interested researchers, PLOS’ Data Policy states that authors cannot be the sole named individuals responsible for ensuring data access (http://journals.plos.org/plosone/s/data-availability#loc-acceptable-data-sharing-methods). Data requests to a non-author institutional point of contact, such as a data access or ethics committee, helps guarantee long term stability and availability of data. Providing interested researchers with a durable point of contact ensures data will be accessible even if an author changes email addresses, institutions, or becomes unavailable to answer requests.

Reply 3: Thank you for pointing out these important requirements. We have revised our Data Availability Statement to address these concerns. Masashi Nagao, at the Clinical Research & Trial Center, Juntendo University, has been designated as the institutional data manager (nagao@juntendo.ac.jp).

Reviewer(s)’ Comments to Author:

Reviewer: #1

Comments to the Author:

I would like to commend the authors for conducting this valuable study and for the opportunity to review it. Developing simple and practical algorithms to predict bacteremia, especially patient-friendly ones, is crucial for reducing sepsis-related mortality and morbidity. This article evaluates the utility of “self-reported” food consumption and shaking chills immediately before taking blood cultures (BC) in predicting the presence or absence of bacteremia in outpatients, primarily those arriving by ambulance. The findings suggest that these self-reported factors are useful predictors of bacteremia in this population. The study’s objectives and clinical implications are clear, and its results hold potential relevance for both patients and practitioners. However, revisions are necessary to enhance the scientific rigor and ensure readers can interpret the findings appropriately.

Reply: Thank you for your insightful comments and for recognizing the value of our study. We appreciate your constructive feedback and have revised the manuscript accordingly to enhance its scientific rigor.

Major comments

Comment 1) Inclusion criteria

The authors assessed the diagnostic accuracy of food consumption and shaking chills separately. It would be helpful to clarify why patients who reported only food consumption or shaking chills were excluded from the analysis. Please provide the rationale for this decision and discuss how it might impact the generalizability of the findings.

Reply 1: Thank you for your insightful comment. We apologize for the insufficient explanation. In our previous study (Komatsu T, et al., J Hosp Med 2017), we observed that some patients with bacteremia maintained normal food consumption, regardless of whether they experienced shaking chills. Based on this finding, we concluded that both self-reported food consumption and shaking chills were necessary to fully assess their combined predictive value in identifying bacteremia.

We have now clarified this rationale in the subsection “Exclusion Criteria” (Page 7, Lines 110–114) of the “Materials and Methods” section. Additionally, we recognize that this exclusion criterion may affect the generalizability of our findings. To address this, we have incorporated a discussion of this limitation by combining the fourth and final limitations into a single revised fourth limitation in the “Discussion” section (Page 29, Line 355–Page 30, Line 363).

Comment 2) Potential confounding

If available, please provide data on the time elapsed between symptom onset and BC collection. Since “food consumption immediately before BC collection” could be influenced by this time interval, understanding its impact on the study results is critical.

Reply 2: Thank you for your insightful comment. As you correctly pointed out, understanding the time interval between symptom onset, the last meal, and blood culture (BC) collection is critical for interpreting the diagnostic accuracy of self-reported food consumption. However, we did not collect this information in the questionnaire, as accurately determining the timing of symptom onset was challenging for many patients.

To address this limitation, we have now clarified this point in the “Materials and Methods” section by adding “(i.e., last meal)” to the “Food Consumption” subsection (Page 9, Line 153).

Comment 3) Impact of predictors on BC collection decisions

Physicians’ awareness of the predictors (e.g., food consumption and shaking chills) could have influenced their decision to collect BC. If this information was accessible to physicians before BC collection, it may introduce bias. To address this, the authors might consider discussing whether the prevalence of bacteremia observed in this study (12.7%) aligns with rates reported in similar settings. This comparison could help assess the potential bias in the study results.

Reply 3: Thank you for your thoughtful comment. Since this study was conducted in a clinical setting, physicians made decisions regarding blood culture (BC) collection based on their usual clinical reasoning, which included patient history, physical examination, and laboratory findings. As a result, physicians were naturally aware of the predictors (food consumption and shaking chills) before deciding whether to collect BCs, which could have introduced a potential selection bias.

To assess the impact of this possible bias, we compared the prevalence of bacteremia in our study (12.7%) with rates observed in similar studies. Notably, our previous study, which used the same decision-making process for BC collection, reported a bacteremia prevalence of 12.0% (221/1,847). Additionally, a study conducted in an emergency department (Rothe K, et al., BMC Infect Dis 2019) found a prevalence of 14.3%. These comparable rates suggest that physician decision-making in our study was consistent with established clinical practice and likely did not introduce significant bias.

We have incorporated these points into the “Collecting Blood Cultures” subsection in the Methods section (Page 7, Lines 118–120), the “Patient Characteristics” subsection in the Results section (Page 12, Line 204), and the Discussion section (Page 29, Lines 345–350), where this issue is addressed as a second limitation.

Comment 4) Multiple logistic regression models

To assess the diagnostic performance of food consumption and shaking chills, it would be appropriate to use multiple logistic regression models that adjust for potential confounding variables. Instead of employing forward or backward stepwise selection, variable inclusion should be guided by a conceptual model outlining plausible confounders. Additionally, given that over 10% of patients were excluded from the analysis due to missing data, the authors should address the potential impact of these exclusions. Performing sensitivity analyses using imputation methods to handle missing data could strengthen the robustness of the findings.

Reply 4: Thank you for your insightful comment. We acknowledge the importance of using a multiple logistic regression model that adjusts for potential confounders based on a conceptual framework. While we initially considered using imputation methods to address missing data, accurately calculating the Systemic Inflammatory Response Syndrome (SIRS) and quick Sequential Organ Failure Assessment (qSOFA) scores proved challenging due to inaccurate data on components that make up the SIRS criteria (body temperature, heart rate, respiratory rate, and white blood cell count) and the qSOFA score (Glasgow Coma Scale, systolic blood pressure, and respiratory rate). Since these variables play a critical role in predicting bacteremia, any imputation might have introduced substantial inaccuracies in classifying patients as having SIRS or qSOFA ≥2. As a result, we opted for a complete-case analysis, acknowledging the trade-off involved.

To clarify our approach, we have added explanations regarding the components of SIRS and qSOFA in the “Other Predictive Variables” subsection (Page 10, Lines 174–177) and the “Statistical Analysis” subsection (Page 11, Lines 189–194) of the Materials and Methods section.

Additionally, we acknowledge that 83 patients were excluded from the multiple logistic regression analysis due to missing vital sign data. To assess the potential impact of these exclusions, we examined the characteristics of these patients. Notably, except for one case, none were transported by ambulance, suggesting that they were likely not in critically ill condition. Therefore, while their exclusion may have led to some degree of selection bias, the overall effect on the association between SIRS/qSOFA and bacteremia is expected to be minimal. We have now included this discussion in the Discussion section (Page 28, Lines 322–329).

Minor comments

Comment 1) Guidelines

Since the article focuses on diagnostic performance, adherence to the STARD guidelines would be more appropriate than STROBE. Please consider revising the reporting accordingly.

Reply 1: Thank you for pointing this out. We acknowledge that the STARD guidelines are more appropriate for reporting diagnostic performance studies than the STROBE guidelines. In response to your suggestion, we have revised the manuscript to align with STARD recommendations where applicable. Specifically, we have updated the “Ethics Approval” subsection (Page 7, Lines 103–104) in the Materials and Methods section to reflect this change. Additionally, we have reviewed the manuscript to ensure adherence to key elements of the STARD checklist, including clear reporting of study design, patient selection, index and reference tests, and diagnostic accuracy measures.

Comment 2) Interrater reliability for the assessment of contamination

The assessment of blood culture contamination is an important aspect of the study. Please provide data on the interrater reliability of this assessment to ensure the consistency and validity of these findings.

Reply 2: Thank you for your comment. We apologize for the insufficient information. In this study, contamination was assessed based on a consensus decision by the infection control team as part of routine medical practice. While formal interrater reliability metrics were not calculated, the determination of contamination followed predefined institutional criteria to ensure consistency. These criteria included the type of organism isolated, clinical correlation with the patient’s presentation, and laboratory findings.

Since there were no disagreements among the infection control team members, calculating interrater reliability was not necessary. However, to clarify this process, we have added more details about the definition and assessment of contamination in the “Definition of True Bacteremia and Contamination” subsection (Page 8, Lines 131–141) of the Materials and Methods section.

Reviewer: #2

Comments to the Author:

Thank you for the opportunity to review this study. The authors have investigated the relationship between dietary intake and the positivity rate of blood cultures. I would like to point out several concerns to improve the quality of this study.

Reply: Thank you very much for your comments and for taking the time to review our manuscript. We appreciate your concerns regarding the relationship between dietary intake and the positivity rate of blood cultures, and we have carefully considered each of the points you raised. We believe that these revisions have improved the quality of our study and addressed the issues you identified. Thank you again for your thoughtful feedback, which has helped us strengthen our manuscript.

Mager comments:

Comment 1) Abstract: Even if a blood culture is a false positive, it is not difficult to determine contamination, and false negatives have a greater impact. Therefore, efforts should be made to reduce false negatives. Poor food consumption had a negative likelihood ratio of 0.66 (95% CI: 0.23-1.94). As the authors wrote it in the limitation, the food consumption was only useful for blood cultures collected between 10 pm and 8 am. A negative predictive value between 8 am and 10 pm was 0.32 (0.12-0.89). This figure was not so low, and it is insufficient to exclude bacteremia.

Reply 1: Thank you very much for your insightful comments. We fully agree that the utility of self-reported normal food consumption as a rule-out criterion during daytime hours (8 am – 10 pm) is limited, as indicated by the negative predictive value (0.32, 95% CI: 0.12–0.89).

In response to this, we have revised both the Abstract (Page 4, Lines 46–47) and the Conclusions (Page 31, Lines 376–378) to better reflect this limitation

Comment 2) Study design: Please list the names of the three hospitals that participated in the study. Are they all university hospitals? The fact that only 715 blood cultures were ordered in two years is significantly lower compared to community hospitals. Are there few patients who visit without an appointment due to fever?

Reply

---

## [Decision Letter · Decision Letter 1]

Dear Dr. Komatsu,

Thank you for submitting your manuscript to PLOS ONE. After careful consideration, we feel that it has merit but does not fully meet PLOS ONE’s publication criteria as it currently stands. Therefore, we invite you to submit a revised version of the manuscript that addresses the points raised during the review process.

We look forward to receiving your revised manuscript.

Kind regards,

Adriana Calderaro

Academic Editor

PLOS ONE

Reviewers' comments:

Reviewer's Responses to Questions

**Comments to the Author**

Reviewer #1: (No Response)

Reviewer #2: All comments have been addressed

2. Is the manuscript technically sound, and do the data support the conclusions?

Reviewer #1: Partly

Reviewer #2: Yes

3. Has the statistical analysis been performed appropriately and rigorously?

Reviewer #1: I Don't Know

Reviewer #2: Yes

4. Have the authors made all data underlying the findings in their manuscript fully available?

Reviewer #1: No

Reviewer #2: Yes

5. Is the manuscript presented in an intelligible fashion and written in standard English?

Reviewer #1: Yes

Reviewer #2: Yes

Reviewer #1: I thank the authors for their thoughtful revision. Most of my previous comments are now addressed; however, I would like the authors to reconsider the rest.

Major comments

Comment 1. Inclusion criteria

I am unsure whether the authors’ reason for including only patients who reported both food consumption and shaking chills is right. The authors said that “we concluded that both self-reported food consumption and shaking chills were necessary to fully assess their combined predictive value in identifying bacteremia,” however, I couldn’t find any combined (food consumption and shaking chills) predictive value in identifying bacteremia.

Comment 2. Multiple logistic regression models

The authors did not fully respond to my previous comments about why they did not include variables based on any conceptual model instead of employing forward or backward stepwise selection.

Reviewer #2: Thank you for revising the manuscript. In Table 2, 'Streptococcus bovis' is listed, but this is the former name, and the current name is 'Streptococcus gallolyticus.' Other than this, there are no particular points that need correction.

**Do you want your identity to be public for this peer review?** For information about this choice, including consent withdrawal, please see our Privacy Policy

Reviewer #1: No

Reviewer #2: **Yes: ** Tomohiro Taniguchi

---

## [Author Response · Author response to Decision Letter 2]

14 May 2025

Reviewer(s)’ Comments to Author:

Reviewer: #1

Comments to the Author:

I thank the authors for their thoughtful revision. Most of my previous comments are now addressed; however, I would like the authors to reconsider the rest.

Reply: We are very grateful for your constructive feedback. In response to your comments, we have revised the manuscript to enhance its scientific rigor

Major comments

Comment 1) Inclusion criteria

I am unsure whether the authors’ reason for including only patients who reported both+ food consumption and shaking chills is right. The authors said that “we concluded that

both self-reported food consumption and shaking chills were necessary to fully assess their combined predictive value in identifying bacteremia,” however, I couldn’t find any combined (food consumption and shaking chills) predictive value in identifying bacteremia.

Reply 1: Thank you for pointing that out. We sincerely regret the confusion. The sentence, “we concluded that both self-reported food consumption and shaking chills were necessary to fully assess their combined predictive value in identifying bacteremia,” refers to our previous study, not the current one. However, we mistakenly used the term “self-reported” instead of “nurse-assessed.” During the proofreading process, we failed to notice that the content had changed to something different from our original intent. We sincerely apologize for not thoroughly reviewing the document. We are very grateful that you pointed this out.

In the present study, we did not intend to assess the combined diagnostic accuracy of these two signs. However, we did believe it was necessary to gather detailed information on patients with bacteremia who had maintained normal food consumption, regardless of whether they exhibited shaking chills. This is because, in a previous study, some bacteremia patients with normal food consumption also had shaking chills, as described in the “Exclusion Criteria” section of the Materials and Methods. Therefore, we required both the food consumption data and the grade of chills.

Comment 2) Multiple logistic regression models

The authors did not fully respond to my previous comments about why they did not include variables based on any conceptual model instead of employing forward or backward stepwise selection.

Reply 2: Thank you for emphasizing the importance of an a priori conceptual framework for covariate selection. In response to your recommendation, we have re-conducted the multiple logistic regression analysis by including the following variables, which were forced into the model based on our conceptual model of bacteremia risk: sex, CRP >10 mg/dL, shaking chills, and SIRS, all of which have been consistently identified in prior studies [1, 12]. Additionally, age ≥75 years, food consumption, qSOFA score ≥2, method of arrival, and timing of blood culture collection were also analyzed as other covariates.

The revised analysis (Table 6, p.27) yields the following ORs (95% CIs): shaking chills 3.20 (1.53–6.67), age ≥75 years 2.89 (1.41–5.90), SIRS 2.75 (1.28–5.54), CRP >10 mg/dL 2.74 (1.50–4.98), and transportation by ambulance 2.46 (1.17–5.18). These estimates differ only marginally and do not materially affect the interpretation; therefore, no changes were required in the Discussion section.

We have revised the “Statistical Analysis” subsection (p. 12, lines 193–197) of the Materials and Methods section to describe the conceptual model approach and have updated the “Ruling in Bacteremia” subsection (p. 25, lines 287–289; p. 26, lines 290–300) of the Results section.

Reviewer: #2

Comments to the Author:

Thank you for revising the manuscript. In Table 2, ‘Streptococcus bovis’ is listed, but this is the former name, and the current name is ‘Streptococcus gallolyticus.’ Other than this, there are no particular points that need correction.

Reply: Thank you for drawing our attention to the current nomenclature. We have replaced “Streptococcus bovis” with “Streptococcus gallolyticus” in Table 2 (p. 16). We have also reviewed the entire manuscript to ensure that the updated name is used consistently and correctly italicized throughout. We appreciate your careful review and trust that this correction meets your expectations.

---

## [Decision Letter · Decision Letter 2]

Diagnostic accuracy of self-reported food consumption and shaking chills in predicting bacteremia in outpatients: A prospective, multicenter observational study

PONE-D-24-58497R2

Dear Dr. Komatsu,

We’re pleased to inform you that your manuscript has been judged scientifically suitable for publication and will be formally accepted for publication once it meets all outstanding technical requirements.

Kind regards,

Adriana Calderaro

Academic Editor

PLOS ONE

Reviewers' comments:

Reviewer's Responses to Questions

**Comments to the Author**

Reviewer #1: All comments have been addressed

Reviewer #2: (No Response)

2. Is the manuscript technically sound, and do the data support the conclusions?

Reviewer #1: Yes

Reviewer #2: (No Response)

3. Has the statistical analysis been performed appropriately and rigorously?

Reviewer #1: Yes

Reviewer #2: (No Response)

4. Have the authors made all data underlying the findings in their manuscript fully available?

Reviewer #1: Yes

Reviewer #2: (No Response)

5. Is the manuscript presented in an intelligible fashion and written in standard English?

Reviewer #1: Yes

Reviewer #2: (No Response)

Reviewer #1: (No Response)

Reviewer #2: (No Response)

**Do you want your identity to be public for this peer review?** For information about this choice, including consent withdrawal, please see our Privacy Policy

Reviewer #1: No

Reviewer #2: **Yes: ** Tomohiro Taniguchi

---

## [Editor Report · Acceptance letter]

PONE-D-24-58497R2

PLOS ONE

Dear Dr. Komatsu,

I'm pleased to inform you that your manuscript has been deemed suitable for publication in PLOS ONE. Congratulations! Your manuscript is now being handed over to our production team.

Kind regards,

on behalf of

MD, PhD, Full Professor Adriana Calderaro

Academic Editor

PLOS ONE